# Assessing Black Locust Biomass Accumulation in Restoration Plantations

Gavriil Spyroglou [1,*][iD], Mariangela Fotelli [1][iD], Nikos Nanos [2][iD] and Kalliopi Radoglou [3][iD]

1 Forest Research Institute, Hellenic Agricultural Organization DEMETER, Vasilika, 57006 Thessaloniki, Greece; fotelli@fri.gr
2 School of Forestry and Natural Environment, Aristotle University of Thessaloniki, University Campus, 54124 Thessaloniki, Greece; nikosnanos@for.auth.gr
3 Department of Forestry and Management of the Environment and Natural Resources, Democritus University of Thrace, Ath. Pandazidou 193 Str., P.O. Box 129, 68200 Orestiada, Greece; kradoglo@fmenr.duth.gr
* Correspondence: spyroglou@fri.gr; Tel.: +30-23-1046-1171 (ext. 233)

**Abstract:** Forests (either natural or planted) play a key role in climate change mitigation due to their huge carbon-storing potential. In the 1980s, the Hellenic Public Power Corporation (HPPC) started the rehabilitation of lignite post-mining areas in Northwest Greece by planting mainly black locust (*Robinia pseudoacacia* L.). Today, these plantations occupy about 2570 ha, but the accumulation of Above Ground Biomass (AGB) and deadwood has not been assessed to date. Therefore, we aimed at estimating these biomass pools by calibrating an allometric model for AGB, performing an inventory for both pools and predicting the spatial distribution of AGB. 214 sample plots of 100 $m^2$ each were set up through systematic sampling in a grid dimension of 500 × 500 m and tree dbh and height were recorded. AGB was estimated using an exponential allometric model and performing inventory measurements and was on average 57.6 t ha$^{-1}$. Kriging analysis reliably estimated mean AGB, but produced errors in the prediction of high and low biomass values, related to the high fragmentation and heterogeneity of the studied area. Mean estimated AGB was low compared with European biomass yield tables for black locust. Similarly, standing deadwood was low (6–10%) and decay degrees were mostly 1 and 2, indicating recent deadwood formation. The overall low biomass accumulation in the studied black locust restoration plantations may be partially attributed to their young age (5–30 years old), but is comparable to that reported in black locust restoration plantation in extremely degraded sites. Thus, black locust successfully adapted to the studied depositions of former mines and its accumulated biomass has the potential to improve the carbon footprint of the region. However, the invasiveness of the species should be considered for future management planning of these restoration plantations.

**Keywords:** climate change mitigation; forest restoration; forest biomass estimation; standing and lying dead wood; variogram model; kriging regression

## 1. Introduction

Forests are considered important components in climate change mitigation strategies for being key drivers of greenhouse gas removals and for facilitating the global community target of net-zero emission by 2050 [1]. Indeed, the Land Use, Land Use Change and Forestry sector (LULUCF) may provide approximately 25% of emission reductions according to nationally determined contributions of the Paris Climate Agreement [2]. A large part of emission reductions is expected from the surplus accumulation of Above Ground Biomass (AGB), a carbon pool that represents 30% of the total carbon of terrestrial ecosystems [1]. In this sense, countries and companies worldwide need to adopt strategies towards enhancing forest carbon sequestration and reducing greenhouse gas emissions [3].

Among forest management actions aiming at maximizing the forest carbon sequestration potential, increasing forest cover in degraded non-forested land is one of the most

promising approaches [4]. Major challenges to this respect include evaluating the growth potential and the associated AGB accumulation of species with high adaptation potential to adverse growth conditions and to changing climate [5]. To this direction, data-based information on growth potential, biomass accumulation and associated carbon assimilation capacity of forest plantations is essential to estimate their climate change mitigation potential and plan their future management.

Black locust is the second most often planted tree species worldwide, after eucalyptus [6]. It is native to North America and has been introduced in Europe at the beginning of the 17th century [7]. In Greece and Europe, black locust is characterized as a naturalized alien plant species [8–10]. According to Sitzia et al. [10], the core area of its distribution is in sub-Mediterranean to warm continental climates, characterized by a high heat-sum. Warmer and drier climate may cause a shift in the distribution limits of black locust (*Robinia pseudoacacia* L.), as it has been described for several native Mediterranean and other European forest species [11]. Among the alien species, black locust was foreseen to increase its distribution area under different climate change scenarios [12,13]. On the contrary, a recent study suggested that, by 2050, a decline of potential niches of this species may take place in Southern Europe [14]. However, black locust exhibits great plasticity of its growth under diverse climatic conditions, indicating a high acclimation potential to future climate changes [15].

There is great concern about the future management of black locust in Europe due to its ecological disadvantages related to its invasiveness, imposed threats for biodiversity and induced changes in light microclimate and soil traits, as discussed in Refs. [16–19]. At the same time, black locust has certain advantages, due to its drought-tolerance [20], tolerance of alkalic soils with pH up to 8 [7], N-fixation ability [21–23], fast growth and high root sucking ability resulting in dense root biomass [24]. Based on these beneficial traits, black locust is often used for the production of timber, firewood, animal forage and for apiculture, as well as for energy plantations [25]. In Greece, black locust has been planted for torrent stabilization on mountains and for soil erosion control on rivers, roads and railway banks. It has been also used as a fodder in silvopastures [26] and as an alternative crop for privately owned plantations, in line with the 2080/92 and 1257/99 European Union (EU) Regulations, due to its great adaptation to marginal agricultural lands [9]. Furthermore, black locust is a suitable non-native species for the restoration of sites extremely degraded by anthropogenic activities such as mining, as it can survive on nutrient-poor depositions like the ones of former open-cast mining areas [20].

The Hellenic Public Power Corporation (HPPC) installed four large power plants in Northwest Greece, close to the largest coal deposit of the country (2.3 billion t), while the open-cast mining activities began in the 1950s to cover the country's increasing electricity demands. In the 1980s, the HPPC started the rehabilitation of the open-cast mining fields in the Lignite Center of Northwest Greece by planting black locust in mixture with other species. To date, the establishment of plantations is still ongoing, while their current area is 2570 ha, making HPPC the largest private forest plantations owner in Greece, owning 26% of the country's black locust plantations area [27]. However, there is neither an inventory for those plantations, nor any allometric equations available for an accurate estimation of their AGB.

The estimation of AGB per forest unit area is a major challenge in the context of carbon stock estimation and dynamics. AGB estimation over a domain is based on a combination of ground reference measurements such as classical field-data (forest inventory data) and the harvest of selected trees that are subsequently oven-dried and weighed for deriving accurate estimation of above ground biomass [28,29]. In some cases, however, auxiliary variables derived from satellite-based products (including LIght Detection And Ranging (LIDAR) technology) are used to render more precise spatial estimation of AGB over the desired domain [30–32]. Ground reference measurements can provide accurate estimates for the calibration and validation of global satellite-based AGB models. In both cases,

geostatistics are applied to estimate the spatial distribution of AGB over a forested area using both data-sources [33–35].

The aims of our study were to (a) calibrate an allometric above ground biomass model for the studied black locust plantations, (b) to provide a reliable estimation of the live and dead above ground biomass distribution across the restored lignite mining areas of Northwest Greece by conducting an inventory of black locust plantations, and (c) to map the AGB pools over the study area by using a geostatistical approach. Our hypotheses were: (1) that the adopted geostatistical analysis would provide a reliable estimation of AGB, in relation to the determined inventory measurements and (2) that the estimated AGB of the studied plantations would indicate the satisfactory development of the restoration at the Lignite Center of Northwest Greece.

## 2. Materials and Methods

### 2.1. Study Area

In the present study, we analyzed the above ground biomass data collected from the black locust restoration plantations of the former open-cast mining areas of the lignite center of Western Greece (Figure 1). The plantations are located near Amyntaio (40.56° to 40.61° N and 21.62° to 21.69° E) and Ptolemaida (40.39° to 40.51° N and 21.7° to 21.89° E), in NW Greece (Figure 1). Black locust covers more than 95% of the planted area, followed by weaver's broom (*Spartium junceum* L) and Arizona cypress (*Cupresus arizonica* Greene), covering 2.45% and 1.44%, respectively. Other planted species comprise oaks, maples, pines and various deciduous broadleaves in very small percentages. The total area of the black locust plantations is approximately 2570 ha, and the elevation ranges from 530 to 950 m. The plantations are established on open-cast mining depositions with varying topography (moderate to steep slopes, plains and terraces). The landscape is fragmented by different land uses, which, apart from the forest plantations, also includes grass lands, agricultural lands and bare lands for photovoltaic parks and recycling facilities. The temperature of the region ranges from 6.1 to 17.4 °C with a mean annual temperature of 12.2 °C, and the total annual precipitation is 664 mm (50 years average values).

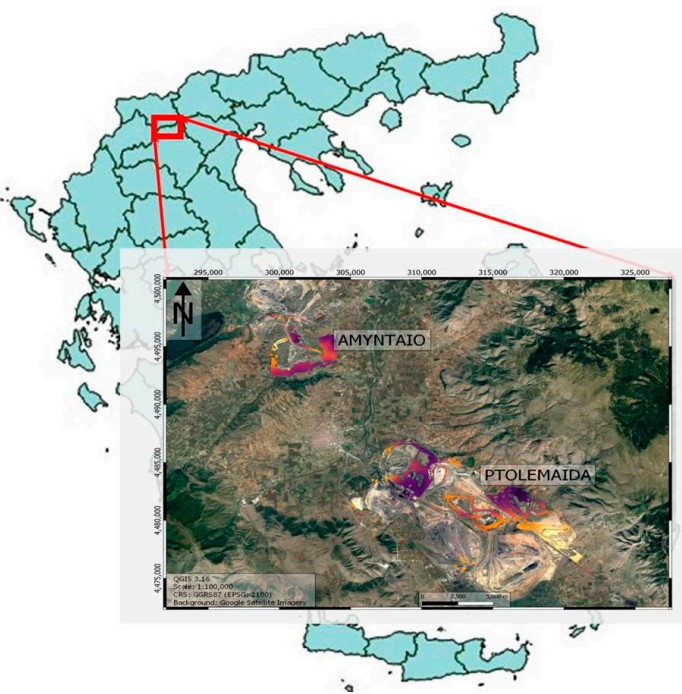

**Figure 1.** Location of the study area in Greece. Colored polygons indicate the studied restoration plantations of black locust in Amyntaio and Ptolemaida mine fields.

### 2.2. Inventory Data

For the estimation of the AGB distribution, in total 214 circular sample plots of 100 m$^2$ each were set up through systematic sampling in a grid dimension of 500 × 500 m in the open-cast mining fields of Amyntaio and Ptolemaida. The field campaign was carried out in the summer of 2019. In each sample plot, the coordinates in the Hellenic Geodetic Reference System (HGSA 87), the tree species, the tree status (dead or alive), the diameter at breast height (*dbh*) in cm using diameter measuring tape, the tree height (*Ht*), and the height to the base of live crown (*Hlc*) in m using Haglöf hypsometer were recorded for all trees. In addition, the lying dead wood (coarse woody debris) defined as dead, woody material of trees with a diameter of >1 cm was recorded for all fallen trunks, fragmented woody branches either lying on the ground or stuck above ground level. The diameter at two ends of the log (*d1* for the small and *d2* for the large, in cm) and the length of the log *L* in m were determined. Diameter measuring tape was used to measure diameters of pieces larger than 5 cm in diameter. A five-decay class system was used to classify the recorded woody debris based on morphology and hardness observed in the field following the methodology of Paletto and Tosi [36]. For samples smaller than 5 cm or rotten ones in decay class 5, only one horizontal measurement was taken by a caliper. The woody debris volume was estimated using the formula of the truncated cone and for the conversion of volume into dry biomass, the wood density reduction according to decay classes [30] was used.

### 2.3. Above Ground Biomass Estimation

The above ground biomass was estimated using the simple exponential allometric model of the form M = a*dbh$^b$. For the calibration of the allometric equation, 30 black locust trees covering all diameter range appeared in the study site were destructively sampled during the summers of 2019 and 2020. The diameter at stump height (D0.3 in cm) and at breast height (dbh in cm) was measured before the tree felling. The sampled trees were cut at the stump height and, after felling, total tree height (H in m), diameter at 50% of bole length (D0.5 in cm) and diameter at the base of live crown were recorded. Each one of the 30 stems was divided into six sections (including the stump). After felling, the fresh weight of each stem section was measured in the field. From each section, a 7 cm wide stem disk was taken, weighed, and oven-dried at the lab at 80 °C until a constant weight was reached to determine the fresh/dry biomass ratio. The fresh biomass of the whole crown was also weighed in the field and oven-dried at the lab at 80 °C until a constant weight was reached. For further details on the biomass estimation methodology, see Zianis et al. [37]. Linear regression was used for the parameterization of the log transformed allometric model.

### 2.4. Statistical Analysis

For the estimation of the spatial distribution of the aboveground biomass, a geostatistical approach was applied. For variogram modeling, the experimental variograms were assessed using the spherical and exponential approach. For the spatial interpolation of the above ground biomass at the points where there was no information, the ordinary kriging method was performed based on a grid size of 50 × 50 m for the predictions of above ground biomass in t ha$^{-1}$ The R programming software (sp library) [38] was used in spatial analysis. For variogram analysis and variogram modeling the gstat and geor libraries were also used [39]. Finally, ordinary kriging regression [40] was applied using nugget, range and partial sill as parameters of the best variogram models. To test the quality of kriging regression, we performed a cross-validation using the leaving-one-out strategy.

## 3. Results

The allometric model for the estimation of tree above ground biomass, after the correction for log transformation bias (e$^{(\mathrm{mse}/2)}$), where mse stands for mean square error, was

$$M = 0.1916 \cdot dbh^{2.2882} \cdot 1.00912$$

While the coefficient of determination ($R^2$) was 0.982, the residual standard error (RSE) was 0.1395 and the mean residual standard error (MSE) was 0.0182. The parameter estimates were statistically significant $F$ (1,28) = 1571, $P < 2.2 \times 10^{-16}$. The residuals were normally distributed (Sapiro–Wilk test: $W = 0.97$, $P = 0.55$). There was no evidence of heterosctasticity, (Breusch–Pagan test: $BP = 0.52$, $df = 1$, $P = 0.469$) and autocorrelation (Durbin–Watson test: $D\text{-}W = 2.51$, $P = 0.921$).

The descriptive statistics, before kriging interpolation, of black locust stands data are presented in Table 1. Generally, black locust stands showed high heterogeneity with dbh ranging from 1.4 to 22.3 cm (mean 7.9 $\pm$ 0.25 cm) and AGB from 0.74 to 274.7 t ha$^{-1}$ (mean 59.9 $\pm$ 2.77 t ha$^{-1}$). At the Amyntaio mine field, the inventoried trees had mean dbh 9.3 $\pm$ 0.31 cm, Ht 11.5 $\pm$ 0.31 m and Hlc 6.2 $\pm$ 0.24 m. The AGB ranged from 29.5 to 206 t ha$^{-1}$ (mean 71.9 $\pm$ 3.73 t ha$^{-1}$). At Ptolemaida, the mean dbh was 7.2 $\pm$ 0.32 cm, Ht 9.0 $\pm$ 0.28 m and Hlc 4.7 $\pm$ 0.22 m. The AGB ranged from 0.74 to 274.7 t ha$^{-1}$ (mean 54.4 $\pm$ 3.56 t ha$^{-1}$).

**Table 1.** Descriptive statistics of data measured in the forest inventory plots.

| Parameter | N (tress ha$^{-1}$) | dbh (cm) | Ht (m) | Hlc (m) | BA [1] (m$^2$ ha$^{-1}$) | AGB$_{inv}$ [2] (t ha$^{-1}$) | AGB$_{krig}$ [3] (t ha$^{-1}$) |
|---|---|---|---|---|---|---|---|
| Amyntaio (788,08 ha, $n_{inv}$ = 65, $n_{krig}$ = 2631) | | | | | | | |
| Mean | 1975 | 9.3 | 11.5 | 6.2 | 14.4 | 71.9 | 67.1 |
| Range | 800–3500 | 6.2–20.9 | 6.1–17.2 | 0.55–9.5 | 6.1–31.9 | 29.5–206.0 | 7.3–120.5 |
| SD | 590.2 | 2.3 | 2.3 | 1.8 | 4.6 | 28.1 | 19.3 |
| SE ($\pm$) | 78.2 | 0.31 | 0.31 | 0.24 | 0.61 | 3.73 | 0.38 |
| RSE (%) | 3.95 | 3.29 | 2.68 | 3.85 | 4.21 | 5.19 | 0.56 |
| Ptolemaida (1782.00 ha, $n_{inv}$ = 149, $n_{krig}$ = 7088) | | | | | | | |
| Mean | 2746 | 7.2 | 9.0 | 4.7 | 11.1 | 54.4 | 48.1 |
| Range | 300–30,078 | 1.4–22.3 | 2.5–16.9 | 0.14–9.9 | 0.25–43.6 | 0.74–274.7 | 17.5–81.3 |
| SD | 2896 | 3.5 | 3.1 | 2.5 | 6.8 | 39.8 | 16.7 |
| SE ($\pm$) | 259.0 | 0.32 | 0.28 | 0.22 | 0.61 | 3.56 | 0.20 |
| RSE (%) | 9.43 | 4.4 | 3.08 | 4.7 | 5.5 | 6.55 | 0.41 |
| Whole study site (2570.08 ha, n = 214) | | | | | | | |
| Mean | 2505 | 7.9 | 9.8 | 5.2 | 12.1 | 59.9 | |
| Range | 300–30,078 | 1.4–22.3 | 2.5–17.2 | 0.14–9.9 | 0.25–43.6 | 0.74–274.7 | |
| SD | 2445.9 | 3.4 | 3.1 | 2.4 | 6.4 | 37.4 | |
| SE ($\pm$) | 181.3 | 0.25 | 0.23 | 0.18 | 0.47 | 2.77 | |
| RSE (%) | 7.23 | 3.16 | 2.35 | 3.42 | 3.90 | 4.63 | |

[1] BA stands for stands for Basal Area, [2] AGB$_{inv}$ stands for above ground biomass from inventory plots, [3] AGBk$_{rig}$ stands for above ground biomass from kriging regression, $n_{inv}$: inventory sample size, $n_{krig}$: kriging sample size, SD: Standard Deviation, SE: Standard Error, RSE: Relative Standard Error.

Due to high biomass heterogeneity in the inventoried plots, the logarithmic transformation of biomass values was used for normality correction. The log transformed values follow a normal distribution with mean m = 4.13 and standard deviation σ = 0.38 for Amyntaio (Figure 2 left) and mean m = 3.53 and standard deviation σ = 1.1 for Ptolemaida (Figure 2 right).

### 3.1. Amyntaio Mine Field

For the estimation of the spatial distribution of the above ground biomass in the Amyntaio mine field using geostatistics, the first step was to analyze and estimate the experimental semivariogram. The exponential, spherical and Gaussian models were fitted to semivariogram data. The spherical model had the better fit with a nugget effect of 423.3, partial sill of 769.2, leveling off at the range of 1500 m (Figure 3).

The data showed a medium spatial autocorrelation with the nugget to total sill ratio being 0.35. The relatively high nugget effect suggests that there is a high measurement error in the data, possibly due to the short scale variation.

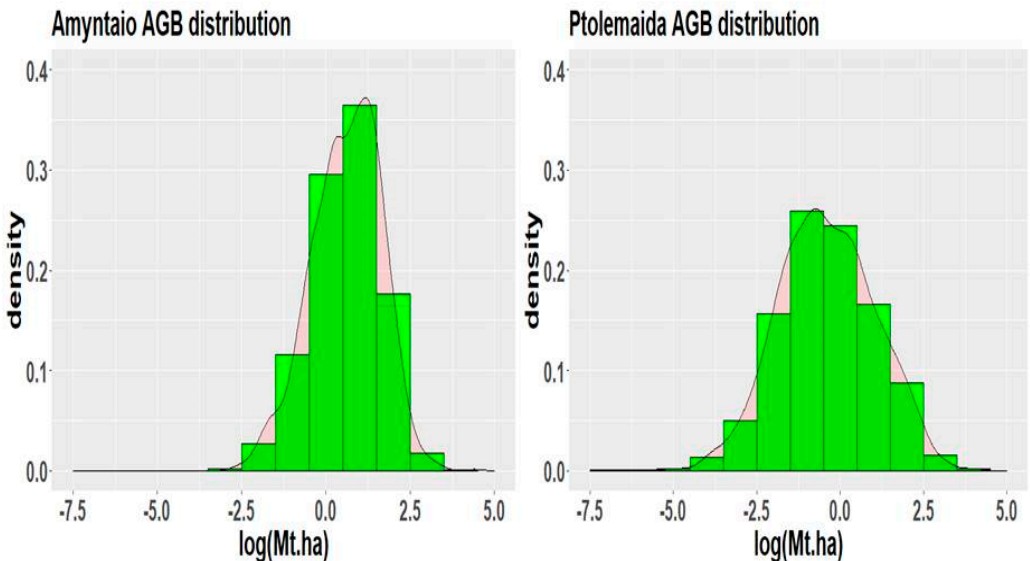

**Figure 2.** Histogram (frequency and density) of the logarithm of AGB (t ha$^{-1}$) at the study sites of Amyntaio (**left**) and Ptolemaida (**right**).

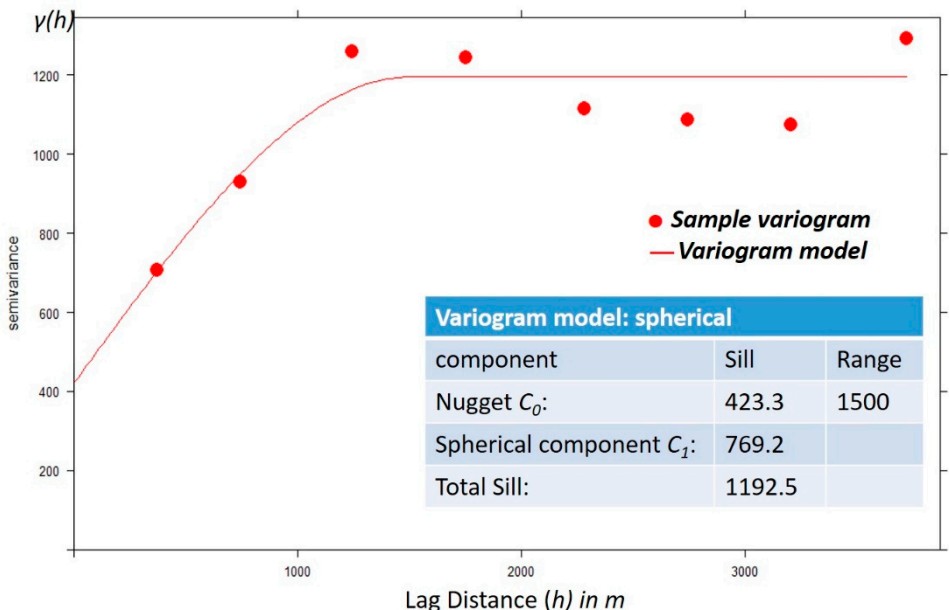

**Figure 3.** Experimental semivariogram with the spherical model fitted at Amyntaio mine area.

The total AGB estimated by ordinary kriging, resulted from a sample of 2631 points based on a raster consisting of $50 \times 50$ m squares, was 52,869.1 t with a mean of $67.1 \pm 0.38$ t ha$^{-1}$. The ABG distribution map derived by the ordinary kriging for Amyntaio mine field is presented in Figure 4.

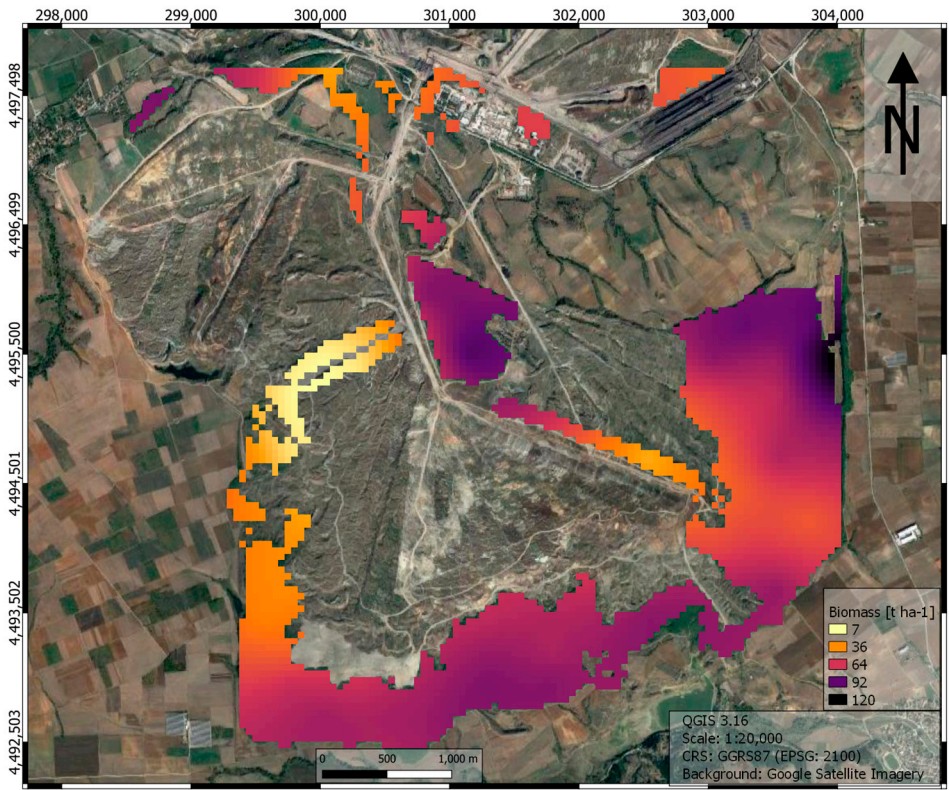

**Figure 4.** Aboveground biomass distribution map at Amyntaio mining area. Coordinates in the Hellenic Geodetic Reference System (HGSA 87).

### 3.2. Ptolemaida Mine Field

For the estimation of the spatial distribution of the above ground biomass in the Ptolemaida mine field using geostatistics, the experimental variogram was also analyzed and estimated from the inventory data. The exponential, spherical and Gaussian models were fitted to semivariagram data. The spherical model was selected because it exhibited a better fit with a nugget effect of 920.4, partial sill of 753.9, leveling off at the range of 5000 m (Figure 5).

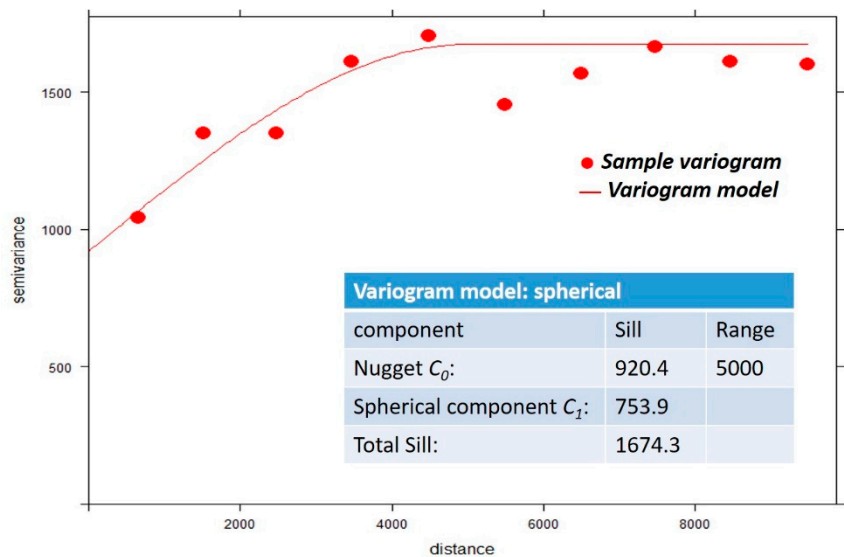

**Figure 5.** Experimental semivariogram with the spherical model fitted at Ptolemaida mine area.

The data showed a similar medium spatial autocorrelation, as found at Amyntaio, given that the nugget to total sill ratio was 0.55. The relatively high nugget effect, higher that the respective nugget effect in Amyntaio, suggests that there was a high measurement error present in the data, possibly due to the short scale variation and to higher degree of fragmentation that is observed in Ptolemaida compared to Amyntaio.

The total AGB estimated by ordinary kriging, resulted from a sample of 7088 points based on a raster consisting of $50 \times 50$ m squares, was 85,772 t with a mean of $48.1 \pm 0.2$ t ha$^{-1}$. The AGB distribution map for Ptolemaida is shown in Figure 6. The kriging estimates of AGB ranged from 17.5 to 81.3 t ha$^{-1}$.

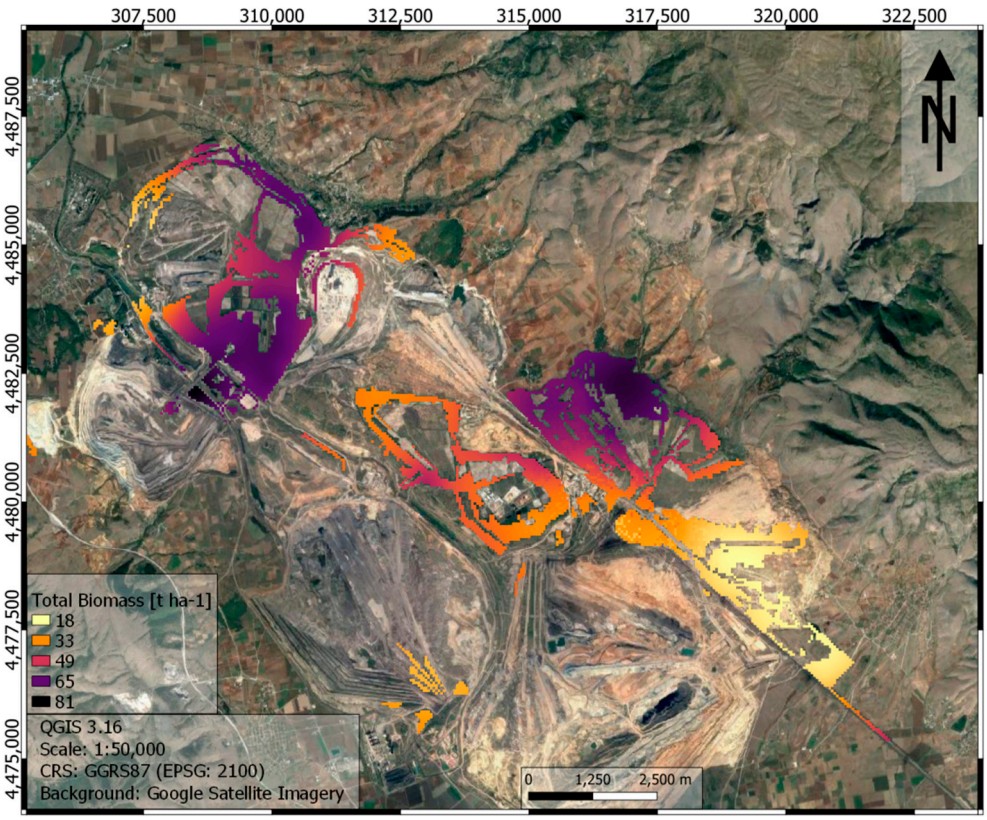

**Figure 6.** Aboveground biomass distribution map at Ptolemaida mining area. Coordinates in the Hellenic Geodetic Reference System (HGSA 87).

The cross-validation of the kriging analysis, which uses the leaving-one-out procedure, is shown in Table 2. The min and max values of kriging predictions were 49.0 and 109.3 t ha$^{-1}$ for Amyntaio and 14.9 and 100.4 t ha$^{-1}$ for Ptolemaida, while the mean observed inventory values were higher and lower, respectively, for both sites. However, the mean AGB estimated by kriging was $75.22 \pm 12.5$ for Amyntaio and $57.34 \pm 20.7$ t ha$^{-1}$ for Ptolemaida and was similar to the observed AGB values, which were $74.84 \pm 29.2$ for Amyntaio and $57.8 \pm 41.3$ t ha$^{-1}$ for Ptolemaida. On average, an error of 28.8 t ha$^{-1}$ of mean AGB can be expected at a given location for Amyntaio and 38.58 t ha$^{-1}$ for Ptolemaida with the application of the krigging analysis.

**Table 2.** Residual statistics of cross-validation results using the leaving-one-out procedure for AGB.

|  | **Min** | **Max** | **Mean** | **SD** | **ME** | **RMSE** |
|---|---|---|---|---|---|---|
| | | | Amyntaio | | | |
| Observed data | 29.5 | 206.0 | 74.84 | 29.2 | | |
| Kriging predictions | 49.0 | 109.3 | 75.22 | 12.5 | −0.383 | 28.77 |
| | | | Ptolemaida | | | |
| Observed data | 0.74 | 275.4 | 57.08 | 41.3 | | |
| Kriging predictions | 14.9 | 100.4 | 57.34 | 20.7 | −0.267 | 38.58 |

SD: Standard Deviation, ME: Mean residual error, RMSE: Root Mean Square Error.

### 3.3. Standing and Lying Dead Wood

The standing dead wood ranged from 0.4 to 17.2 t ha$^{-1}$ for Amyntaio, and 0.08 to 26.8 t ha$^{-1}$ for Ptolemaida mine field respectively (Table 3). The lying dead wood ranged from 0.5 to 19.4 and from 0.5 to 66 m$^3$ ha$^{-1}$.

**Table 3.** Standing and lying black locust dead wood data at the study site.

| Parameter | Lying Dead Wood [1] (m$^3$ ha$^{-1}$) | Lying Dead Wood [2] (t ha$^{-1}$) | Standing Dead Wood (t ha$^{-1}$) | Live Wood (t ha$^{-1}$) |
|---|---|---|---|---|
| | | Amyntaio (n = 65) | | |
| Mean | 5.2 | 1.6 (2.2%) | 4.4 (6.2%) | 71.9 |
| Range | 0.5–19.4 | 0.14 (0.5%)–6.9 (3.3%) | 0.40 (1.4%)–17.2 (8.3%) | 29.5–206.0 |
| SD | 5.1 | 1.7 | 4.0 | 28.1 |
| SE (±) | 1.0 | 0.33 | 0.65 | 3.7 |
| RSE (%) | 19.4 | 20.4 | 14.7 | 5.2 |
| | | Ptolemaida (n = 149) | | |
| Mean | 9.3 | 2.9 (5.3%) | 4.5 (8.3%) | 54.4 |
| Range | 0.5–88.2 | 0.16 (22.8%)–28.2 (10.3%) | 0.1 (14.3%)–26.8 (9.8%) | 0.7–274.7 |
| SD | 15.2 | 5.1 | 4.7 | 39.9 |
| SE (±) | 1.9 | 0.65 | 0.54 | 3.6 |
| RSE (%) | 20.9 | 22.1 | 12.2 | 6.6 |
| | | Whole study site (n = 214) | | |
| Mean | 8.1 | 2.6 (4.3%) | 4.4 (7.3%) | 59.9 |
| Range | 0.5–88.2 | 0.14 (20.0%)–28.2 (10.3%) | 0.1 (14.3%)–26.8 (9.8%) | 0.7–274.7 |
| SD | 13.1 | 4.4 | 4.5 | 37.4 |
| SE (±) | 1.4 | 0.47 | 0.42 | 2.8 |
| RSE (%) | 17.4 | 18.4 | 9.5 | 4.6 |

[1] wood volume m$^3$ ha$^{-1}$, [2] wood dry weight t ha$^{-1}$; n: sample size, SD: Standard Deviation, SE: Standard Error, RSE: Relative Standard Error.

The overall decay degree in the quality scale from 1 to 5 was ranged as: 10% for 1, 27% for 2, 45% for 3, 17% for 4 and 1% for 5 (Figure 7). Eighty five percent of the dead wood concentrates in decay classes 1 and 2 meaning an increase in mortality rate in the last 10 years.

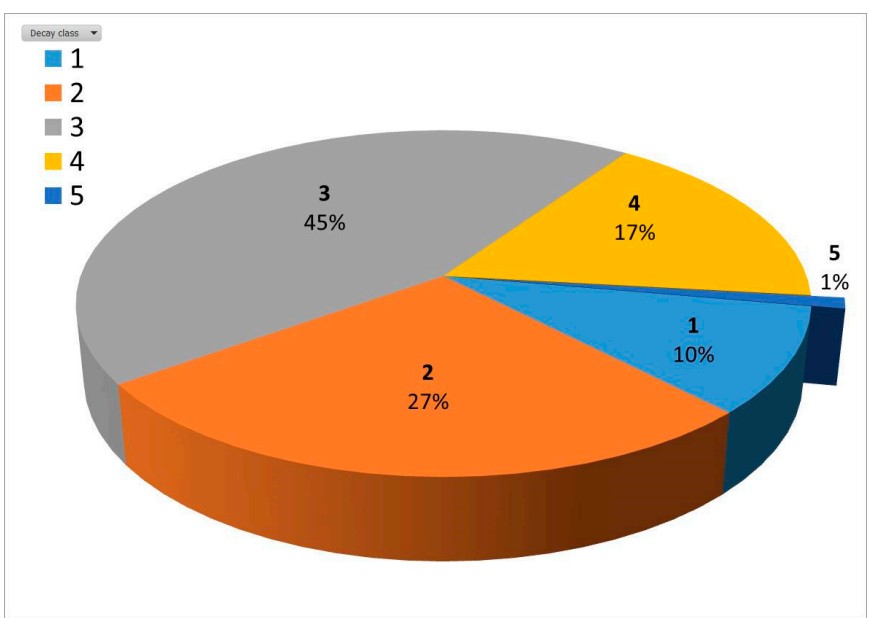

**Figure 7.** Overall distribution of decay classes of dead standing and lying wood in the restored open-cast lignite mining areas.

## 4. Discussion

Black locust is an alien forest species in Europe with controversial traits; it is an invasive species, threatening biodiversity and causing alterations in microclimate and soil conditions in forest plantations e.g., [19,25]. However, its fast growth potential, dense root system due to extensive root suckers [24] and N-fixation ability [35] allow it to survive, adapt and form dense stands under unfavorable conditions. Owing to these features, black locust has been extensively used by the HPPC for the restoration of open-cast mines at the Lignite Centers in Greece, aiming at improving the carbon footprint of these areas by increasing carbon sequestration, while also providing goods and ecosystem services such as fuel, timber, non-wood forest products and recreational opportunities to local populations. The AGB of black locust restoration plantations and its spatial distribution at the Lignite Center of Northwest Greece, established on post-mining depositions, was estimated by the development of an allometric model and the performance of inventory measurements and kriging analysis.

The spatial biomass distribution showed a tendency to increase from Southeast to Northwest in Ptolemaida and from West to North in the Amyntaio mine field, verifying the spatial direction of the planting process in the past years. The semi-variograms of above ground biomass in both Amyntaio and Ptolemaida mine fields were effectively described by the spherical model, although the exponential model has also been selected for broadleaf forests [35] and other ecosystems [41]. Still, the spherical model is one of the most widely applied in ecological studies [40]. The partial sill to total sill ratio was higher than that of nugget to total sill, verifying that the spatial variation of AGB was mainly affected by structural factors, such as the age of the plantations and competition among trees. However, the effects of random factors, such as the sampling resolution and inventory sample plot size cannot be excluded. In Ptolemaida, both structural and random factors equally affected the spatial distribution of AGB. The magnitude of the spatial correlation of AGB was medium at both field sites, with nugget to total sill ratio varying between 25 and 75%. However, the spatial autocorrelation range differed between the two sites. It was 1500 m in Ptolemaida and 5000 m in Amyntaio, indicating that the AGB at a given point affected the AGB of another point in a much lower radius in Amyntaio than in Ptolemaida. The observed high nugget effects could be partially due to the fact that the allometric equation was calibrated from a relatively small number of sampled trees, 30 in our case. Moreover, an increased sampling resolution of the systematic grid



at the inventory field campaign, thus a higher number of sampling plots, may decrease the nugget effects, e.g., as seen in Ref. [42]. However, similarly high nugget effect and spatial autocorrelation range observed at the spatial prediction of forest biomass by Lamsal et al. [43] were attributed to the heterogenous studied landscape, which also applies in our case due to the fragmented and heavily impacted post-mining terrain.

The predicted mean AGB by kriging regression was similar to that measured by the inventory (Table 2). However, low values were overestimated, and high ones were underestimated, leading to a smoother spatial distribution of AGB in the study area. The heterogeneity and fragmentation of the post-mining landscape, together with the above-mentioned potential limitations, associated with the nugget effects and the resolution of the systematic grid, probably explain this difference in AGB estimation between the kriging and the inventory analysis. Jaja et al. [44] and Akhawan et al. [45] similarly concluded that kriging did not produce accurate estimations of AGB because of the high spatial heterogeneity and uneven age of the forests they studied, contrary to other reports [41]. Thus, our first hypothesis regarding the reliable estimation of AGB at the black locust plantations of the Lignite Center of Northwest Greece by means of kriging analysis is only partially confirmed.

The inventory revealed that the growth traits of the plantations varied greatly, due their different age (c. 5–35 years old). Tree density, dbh and height over the entire study area ranged from 300 to 30,078 trees ha$^{-1}$, 1.4 to 22.3 cm and 2.5 to 17.2 m, respectively. Moreover, total biomass fluctuated from 7.3 to 120.5 t ha$^{-1}$. Although the restoration was initiated at Ptolemaida, planting is still active there resulting in many young plantations and, thus, a lower mean dbh and height and a higher tree density (6.9 cm, 9.1 m, and 2746 trees ha$^{-1}$ respectively), than at Amyntaio (9.3 cm, 11.5 m and 1975 trees ha$^{-1}$, respectively). Nicolescu et al. [6] reported that black locust can reach 14 m height in 10 years and dbh up to 20 cm in 25 to 30 years, under optimal conditions. Thus, the rather stressful conditions that the studied black locust plantations cope with explains their lower growth. In fact, our results correspond to the V$^{th}$ growth class, the first of the two poorest classes, of the Hungarian black locust yield and biomass tables [46]. Similarly low was the above ground biomass estimated by Wang et al. [47] for 5 to 25 years old black locust restoration plantations on degraded agricultural lands at the Loess Plateau in China, which ranged from 4.1 to 27.5 t ha$^{-1}$

Deadwood represented 6 to 10% of the standing biomass at the studied plantations. Under unfavorable conditions, black locust cannot withstand shade, resulting in high self-thinning [6]; thus, the amount of dead wood is expected to increase in the future, considering that the studied plantations have not been managed and thinned to date. Consistently, deadwood accounted for 16% of the total carbon stock in 60 years old black locust restoration plantations in Serbia [48]. Moreover, 85% of the estimated dead wood in our study belonged to the decay degrees 1 and 2, indicating that the decay and thus tree mortality was initiated recently, as a result of self-thinning induced by competition for light and other resources.

Thus, the inventory analysis revealed that both the AGB and the deadwood biomass of the studied black locust restoration plantations were relatively low, compared to reported values under optimal conditions. This was both due to their young age and the harsh conditions of the post-mining depositions where the plantations grow. Still, as above-mentioned, the AGB of the studied plantations was comparable to other black locust restored sites developed at rather degraded sites.

Exotic species have been often used for the rehabilitation of bare coal mines due to their successful establishment and fast growth [49,50]. In this context, black locust has been planted for the rehabilitation of heavily degraded sites [23], such as the depositions of former coal mines [16,51] and combustion waste disposal sites [52]. Kraszkiewicz (2021) [53] concludes that planting black locust on various sites of wastelands establishes tree stands for medium size timber production, which perform better that other species such as poplar and willows. Filcheva et al. [50] confirmed the positive impact of afforestation on the initial



soil-forming processes in coalmine spoils classifying black locust as being more beneficial than the black pine because it promotes less acidification, fixes nitrogen and incorporates more organic material into the soil. Thus, the morphological characteristics and the eco-physiological adaptations that the black locust exhibits allow it to grow successfully at extremely degraded post-mining sites [20,50]. However, the dangers that this invasive species imposes for biodiversity indicate that it should be used only for such restoration cases and point to the need for its proper future management.

## 5. Conclusions

Kriging analysis accurately estimated the mean AGB at the black locust restoration plantations of the Lignite Center of Northwest Greece, but presented large errors in the prediction of high and low biomass values. The inventory measurements estimated relatively low, but comparable tree biometric traits and total AGB, as in other black locust restoration plantations in heavily degraded regions. Deadwood biomass was also low, but is expected to increase if the plantations remain out of management in the future. Our results indicate that black locust plantations are successfully established at the studied coal mine spoils, and the produced biomass may contribute to the improvement of carbon footprint, to carbon storage and, thus, to climate change mitigation in such a degraded post-mining area. However, management measures should be taken to limit potential threats related to the invasiveness of the species.

**Author Contributions:** Conceptualization, G.S. and N.N.; methodology, G.S. and N.N.; software, G.S. and N.N.; validation, M.F., N.N.; formal analysis, G.S. and N.N.; investigation, G.S.; data curation, G.S. and N.N.; writing—original draft preparation, G.S. and M.F.; writing—review and editing, G.S.; M.F. and N.N.; supervision, K.R.; project administration, K.R.; funding acquisition, K.R. All authors have read and agreed to the published version of the manuscript.

**Funding:** This research was funded by Single RTDI state Aid Action Research–Create–Innovation with the co-financial of Greece and the European Union (European Regional Development Fund) in context with Operational Program Competitiveness, Entrepreneurship and Innovation (ΕΠΑΝΕΚ) of the NSRF 2014–2020 (project contribution of the tree planted land of West Macedonia lignite center to protection of environment and to mitigation of climate change T1EDK-02521).

**Institutional Review Board Statement:** Not applicable.

**Informed Consent Statement:** Not applicable.

**Data Availability Statement:** Not applicable.

**Acknowledgments:** The authors would like to acknowledge the Hellenic Public Power Corporation (HPPC S.A.) for its substantial contribution with the necessary staff and machinery in order to conduct the field campaigns and data collection. Special thanks are due to Marina Tentsoglidou, Aris Azas, Christos Papadopoulos and their technical team. Special thanks are also due to Stamatis Tziaferidis, ENA Development Consultants, for his valuable help in inventory and tree logging.

**Conflicts of Interest:** The authors declare no conflict of interest.

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
