# Peer review of "Assessing Black Locust Biomass Accumulation in Restoration Plantations"

_forests, doi:10.3390/f12111477_

Round 1

Reviewer 1 Report

Dear Author(s),

I would like to state that the investigation is based on sufficient amount of material; methods are presented clearly enough to allow other researchers to understand the procedures and to repeat the study. In addition, there is an evident response of the results to aim of the manuscript, but some clarifications and corrections are necessary. 

In general, I consider this manuscript to be easy to follow and quite interesting. I only have a few relatively minor comments. I would have liked to see more information concerning some statistics which colud help to estimate od biomass of Black locust stands in the studied area in more appropriate way. For example, in the text there is no data about the area size (ha) for each region separately (Amyntaio and Ptolemaida). To better assess and understand the estimated amount of biomass it is necessary to state the sample size and the number of sampling units for each region separately. Additionally, it could be very useful scientifically and methodologically to calculate the absolute and relative error of biomass assessment for each region separately and for the whole area. Also, it is necessary to compare the average value of aboveground biomass calculated by sampling (with + - allowed error) with the average value calculated by kriging (table) and see if the mean value of aboveground biomass calculated by kriging is within the limits (between upper and lower limit) of estimated aboveground biomass for each region separately and for the whole area. Such results need to be commented on. Further, it is necessary to harmonize data from Figure 3 with data in the text in the Results. Also, some adequate statistics to explain data in Table 3 in more precise way are missing etc.

In the attached file I provide specific minor suggestions that maybe could help you improving the manuscript.

All the best!

Author Response

Dear Author(s),

I would like to state that the investigation is based on sufficient amount of material; methods are presented clearly enough to allow other researchers to understand the procedures and to repeat the study. In addition, there is an evident response of the results to aim of the manuscript, but some clarifications and corrections are necessary. 

In general, I consider this manuscript to be easy to follow and quite interesting. I only have a few relatively minor comments.

 I would have liked to see more information concerning some statistics which could help to estimate od biomass of Black locust stands in the studied area in more appropriate way. For example, in the text there is no data about the area size (ha) for each region separately (Amyntaio and Ptolemaida).

Answer: We’ve added in the manuscript the respected data for each region separately and for the whole site totally.

To better assess and understand the estimated amount of biomass it is necessary to state the sample size and the number of sampling units for each region separately.

Answer: We’ve added in the text the sample size for Amyntaio and Ptolemaida separately

Additionally, it could be very useful scientifically and methodologically to calculate the absolute and relative error of biomass assessment for each region separately and for the whole area.

Answer: We’ve calculated and added in the text and in table 2 the standard error and the relative standard error for each region and in whole study site

Also, it is necessary to compare the average value of aboveground biomass calculated by sampling (with + - allowed error) with the average value calculated by kriging (table) and see if the mean value of aboveground biomass calculated by kriging is within the limits (between upper and lower limit) of estimated aboveground biomass for each region separately and for the whole area. Such results need to be commented on. 

Answer: We’ve carried out cross-validation and present the results in a new table, (table 4) containing all the information suggested by the reviewer and we analyse in the text the results of cross-validation.

Further, it is necessary to harmonize data from Figure 3 with data in the text in the Results. Also, some adequate statistics to explain data in Table 3 in more precise way are missing etc.

Answer: We’ve elaborated the data from Figures 3 and 5 and made all the necessary changes in the results section. Moreover, we’ve added the statistics proposed by the reviewer in table 3 now table 4.

In the attached file I provide specific minor suggestions that maybe could help you improving the manuscript.

All the best!

Line 47 – The right part of bracket number 4 is missing .

Answer: We fixed it

Line 144 – What are the model parameters? There is no applied statistics (for example, coefficient of determination, significance of the parameters, residual analysis).

Answer: We have included text describing model parameters, R square value, significance level and residual analysis. We’ve also reported tests for residual normality (Sapiro Wilks), heteroscedasticity (Breusch –Pagan), and autocorrelation (Durbin- Watson)

Line 111 – Put a space behind 950

Answer: We fixed it

Line 135 – No brackets in front and behind of Paletto & Tosi

Answer: We fixed it

Ine 140 – In Table 1 put a space in 3cm

Answer: We fixed it

Line 163 and 164 – The sentence is not completely clear!

Answer: We have revised the sentence as follows: The R programming software (sp library) [32] was used in spatial analysis. For variogram analysis and variogram modelling the gstat and geor libraries where also used [33]. Finally, ordinary kriging regression [34] was applied using nugget, range and partial sill as parameters of the best variogram models.

Line 169 – If I correctly understood this, it should be put ‘’mean’’ in front of dbh, at end of the line.

Answer: We’ve added the word mean in parenthesis for dbh and above ground biomass including standard error

Line 169 – It is stated that the investigated area has a high heterogeneity of stands structure in terms of diameter and biomass amount, which as a rule requires a larger sample in relation to homogeneous stands. Therefore, it is necessary to determine the level of error in the assessment of biomass (in absolute and relative terms) and compare to the results calculated using kriging procedure.

Answer: We’ve elaborated the reviewer’s suggestion. In table 2 we’ve added: the standard error and the relative standard error for all variables as well as the sample size and site area in ha for each site separately and in total. Moreover, we’ve added another table with the estimated AGB by kriging regression with all respective statistics and compared with the observed biomass values from the inventory

Line175 – The table does not contain the size of the total area and the size of the sample for each part of the area separately (Amyntaio and Ptolemaida). The error of biomass estimation in absolute and relative values is not stated.

Answer: We’ve added the area per site and in whole study area as well as the sample size and the biomass estimation standard error in absolute and relative values

Line 179 and 180 – put a space in m=4.13 and m=3.53

Answer: We’ve put the spaces

Line 196 and 219 – The biomass calculated by kriging was not compared with the biomass calculated by method of the representative (mean values).

Answer: We’ve made the comparisons see also the answer for comment in line 169

Line 230 – There is no appropriate statistics in Table 3 (f.e. standard deviation, the relative standard error of the estimated average wood biomass per hectare etc.)

Answer: We’ve added the appropriate statistics in table 3 now table 4

Line 232 and 233 – The stated data do not correspond to the data in Graph 7!

Answer: The figure was wrong; we’ve replaced it with the correct figure and the data in the text correspond to the data in the figure 7

Line 263 – Mean dbh or dbh?

Answer: It concerns mean values.

 We’ve rephrase the passage as follows: Although the restoration was initiated at Ptolemaida, planting is still active there resulting in many young aged plantations and thus exhibiting mean values of dbh, height and tree density at 6.9 cm, 9.1 m, and 2,746 trees ha-1 respectively, lower than those appeared at Amyntaio site where the mean dbh, height and tree density were (9.3 cm, 11.5 m and 1,975 trees ha-1, respectively).

Line 265 – Put a space behind 9.8

Answer: We’ve put the space

Line 266 – If we look more closely at the tables from Hungary (Redei et.al. 2017), using the rating by the author(s), it can be concluded that the mentioned area on average corresponds more to V, not to VI site (growth) class. As the author stated, the VI site class has an average height of 10.7 m and a diameter of 10.4 cm with 1.429 trees per hectare at the age of 25, which is far above the average height of 9.8 m, 7.5 cm in diameter and below the average 2.762 trees per hectare in the study area (Table 2). On the other hand, V site class (Redei et.al. 2017) at the age of 15 has a height of 10 meters and 2.198 trees per hectare, with a diameter of 8.8 cm, which is much closer to the data from the researched area.

Answer: We agree with the reviewer’s comment, Our mean values of the whole study site are closer to the site class V and age class 15 years of  (Redei et.al. 2017) yield tables that the VI site class and 25 years stated by us in the text. We’ve accepted the reviewer’s suggestion

Line 268 (285) – Please, correct bh 10.4cm

Answer: We’ve corrected it

Line 269 – put a space behind 51.9

Answer: We’ve put the space

Line 275 – It may not be completely methodologically efficient to compare the data from this paper with the data from Germany, due to the different age and site productivity (site classes). According to the data from Germany, the stand aged 8 years reached a biomass of 37.1 t /ha with a height and diameter of 5.6 m and 4.6 cm, respectively.

Answer: Yes we agree that it is not efficient to compare short rotation black locust coppice plantations with our study area because the silvicultural aim is different. The common feature between the study site of publication [38] and our study is both concern black locust plantations planted in post mining areas. For this reason we’ve cited this publication

Line 283 – stablishes?

Answer: We’ve changed to estabishes

Line 390 – Delete number 2019

We’ve corrected the reference

Line 436  - Reference 26 should be corrected.

Answer: We’ve corrected the reference

Thank you very much for your valuable comments and suggestions that helped substantially in improving this manuscript

Reviewer 2 Report

Review of the submitted manuscript entitled Assessing black locust biomass accumulation in restoration plantations.

Robinia pseudoacacia is one of the most widely cultivated alien tree species in Europe and is undoubtedly very useful for the rehabilitation of degraded areas. The aim of this study was "to provide reliable estimation of the live and dead above ground biomass distribution across the restored open-cast Lignite mining areas of Western Greece by conducting an inventory on black locust plantations, calibrating an allometric above ground biomass model and using a geostatistical approach for mapping carbon pools over the study area ".

The goal seems to be consistently achieved, but there is no scientific hypothesis. The materials and methods, and results are well described. More extensive changes are needed in Introduction and Discussion. In some fragments of these chapters, the authors should refer to newer bioclimatological end ecological on the black locus in Europe. 

L.57-61 and L.283-289: Please note that recent studies on the potential distribution of the same authors (refers to reference 10), using updated climatic data and the occurrence date of the investigation by the same authors, shows a slightly different result, see Puchałka et al. (2021). According to this study, Robinia pseudoacacia will lose more areas to the south for its occurrence than earlier studies by Dyderski et al. [10]. If this scenario is fulfilled, only small areas in this country will be optimal for this species. These changes are expected around 2050 (Puchałka et al. 2021). On the other hand, studies by Klisz et al. (2021) show that this species can adapt very plastically to climatic conditions through growth plasticity. Much certainly depends on the soil in which the black locust is planted. Hence, it is necessary to discuss the prospects and limitations of R. pseudoacacia cultivation in the coming decades. 

Puchałka R, Dyderski MK, Vítková M, Sádlo J, Klisz M, Netsvetov M, Prokopuk Y, Matisons R, Mionskowski M, Wojda T, Koprowski M & Jagodziński AM (2021) Black locust (Robinia pseudoacacia L.) range contraction and expansion in Europe under changing climate. Global Change Biology 8:1587–1600.

Klisz M, Puchałka R, Netsvetov M, Prokopuk Y, Vítková M, Sádlo J, Matisons R, Mionskowski M, Chakraborty D, Olszewski P, Wojda T & Koprowski M (2021) Variability in climate-growth reaction of Robinia pseudoacacia in Eastern Europe indicates potential for acclimatisation to future climate. Forest Ecology and Management 492.

The Discussion is a weak part of this manuscript.

L.253-261: This passage does not sound like a Discussion. This is a repetition of more essential results.

L.285: Dydersky -> Dyderski

L.295-296: Recent studies by Puchałka et al. (2021) show that the black locust will lose potential niches in many places in southern Europe. 

Hence, If the scenario starts, other alternatives will have to be found.

Also, It should be mentioned that the black locust is a species considered to be particularly harmful to the biodiversity of European plant communities. I believe that showing only the positive sides of Robinia cultivation is very controversial and may arouse reluctance to this research among many readers. Robinia causes a strong unification of the phylogenetic structure of vegetation and floristic uniformity. Native species decline and are replaced with ruderal plants, see i.e.

Piwczyński, M., Puchałka, R., & Ulrich, W. (2016). Influence of tree plan- tations on the phylogenetic structure of understory plant commu- nities. Forest Ecology and Management, 376, 231–237. https://doi. org/10.1016/j.foreco.2016.06.011  

Slabejová, D., Bacigál, T., Hegedüšová, K., Májeková, J., Medvecká, J., Mikulová, K., Šibíková, M., Škodová, I., Zaliberová, M., & Jarolímek, I. (2019). Comparison of the understory vegetation of native forests and adjacent Robinia pseudoacacia plantations in the Carpathian- Pannonian region. Forest Ecology and Management, 439, 28–40. https://doi.org/10.1016/j.foreco.2019.02.039

Vítková, M., Tonika, J., & Müllerová, J. (2015). Black locust – Successful invader of a wide range of soil conditions. Science of the Total Environment, 505, 315–328. https://doi.org/10.1016/j.scito tenv. 2014.09.104

Vítková, M., Sádlo, J., Roleček, J., Petřík, P., Sitzia, T., Müllerová, J., & Pyšek, P. (2020). Robinia pseudoacacia dominating vegetation types of Southern Europe: Species composition, history, distribution and management. Science of the Total Environment, 707, 134857. https:// doi.org/10.1016/j.scito tenv.2019.134857

The above studies show that sometimes more harm than good from the cultivation of black locust and its introduction should be limited only to justified cases, such as land undergoing remediation taking into account the problems related to the management of biological invasions, see i.e.

Brundu, G., Pauchard, A., Pyšek, P., Pergl, J., Brunori, A., Canavan, S., Campagnaro, T., Celesti-Grapow, L., de Sá Dechoum, M., Dufour-Dror, J.-M., Essl, F., Flory, S. L., Genovesi, P., Guarino, F., Guangzhe, L., Hulme, P. E., Jäger, H., Kettle, C. J., Krumm, F., & Richardson, D. (2020). Global guidelines for the use of non-native trees: Awareness, sustainable use, invasion risk prevention and mitigation. NeoBiota, 61, 65–116. https://doi.org/10.3897/neobi ota.65.58380

L.302-345: Almost the entire Discussion page contains only three references to just two publications. It is simply an interpretation of the results with some conclusions, not a proper scientific discussion.

The full scientific name of the species with authority Robinia pseudoacacia L. should only be given in the text when it is mentioned for the first time. The abbreviated version of R. pseudoacacia should be used each subsequent time.

Kind regards

Author Response

Αρχή φόρμας

Open Review

English language and style

( ) Extensive editing of English language and style required
( ) Moderate English changes required
( ) English language and style are fine/minor spell check required
(x) I don't feel qualified to judge about the English language and style

Yes

Can be improved

Must be improved

Not applicable

Does the introduction provide sufficient background and include all relevant references?

( )

( )

(x)

( )

Is the research design appropriate?

(x)

( )

( )

( )

Are the methods adequately described?

(x)

( )

( )

( )

Are the results clearly presented?

(x)

( )

( )

( )

Are the conclusions supported by the results?

( )

( )

(x)

( )

Comments and Suggestions for Authors

Review of the submitted manuscript entitled Assessing black locust biomass accumulation in restoration plantations.

 Robinia pseudoacacia is one of the most widely cultivated alien tree species in Europe and is undoubtedly very useful for the rehabilitation of degraded areas. The aim of this study was "to provide reliable estimation of the live and dead above ground biomass distribution across the restored open-cast Lignite mining areas of Western Greece by conducting an inventory on black locust plantations, calibrating an allometric above ground biomass model and using a geostatistical approach for mapping carbon pools over the study area ".

The goal seems to be consistently achieved, but there is no scientific hypothesis. The materials and methods, and results are well described. More extensive changes are needed in Introduction and Discussion. In some fragments of these chapters, the authors should refer to newer bioclimatological end ecological on the black locus in Europe. 

 Answer: We have revised the aims of the study to state them more clearly and we have included the hypotheses (please see now Ln 136 – 147).

L.57-61 and L.283-289: Please note that recent studies on the potential distribution of the same authors (refers to reference 10), using updated climatic data and the occurrence date of the investigation by the same authors, shows a slightly different result, see Puchałka et al. (2021). According to this study, Robinia pseudoacacia will lose more areas to the south for its occurrence than earlier studies by Dyderski et al. [10]. If this scenario is fulfilled, only small areas in this country will be optimal for this species. These changes are expected around 2050 (Puchałka et al. 2021). On the other hand, studies by Klisz et al. (2021) show that this species can adapt very plastically to climatic conditions through growth plasticity. Much certainly depends on the soil in which the black locust is planted. Hence, it is necessary to discuss the prospects and limitations of R. pseudoacacia cultivation in the coming decades. 

Puchałka R, Dyderski MK, Vítková M, Sádlo J, Klisz M, Netsvetov M, Prokopuk Y, Matisons R, Mionskowski M, Wojda T, Koprowski M & Jagodziński AM (2021) Black locust (Robinia pseudoacacia L.) range contraction and expansion in Europe under changing climate. Global Change Biology 8:1587–1600.

Klisz M, Puchałka R, Netsvetov M, Prokopuk Y, Vítková M, Sádlo J, Matisons R, Mionskowski M, Chakraborty D, Olszewski P, Wojda T & Koprowski M (2021) Variability in climate-growth reaction of Robinia pseudoacacia in Eastern Europe indicates potential for acclimatisation to future climate. Forest Ecology and Management 492.

Answer: Thank you for the interesting, suggested papers. We have included both the studies of Puchałka et al. (2021) and Klisz et al. (2021), as well as other referring to the properties of black locust and revised the text accordingly

 The prospects and limitations of black locust cultivation in the near future are discussed now in the revised Discussion.

 The Discussion is a weak part of this manuscript.

L.253-261: This passage does not sound like a Discussion. This is a repetition of more essential results.

Answer: The entire Discussion has been revised, to address our hypotheses, include more references and indicate the strengths and weaknesses (and limitations) in the use of black locust.

L.285: Dydersky -> Dyderski

Answer: Done

L.295-296: Recent studies by Puchałka et al. (2021) show that the black locust will lose potential niches in many places in southern Europe. 

 Hence, If the scenario starts, other alternatives will have to be found.

Also, It should be mentioned that the black locust is a species considered to be particularly harmful to the biodiversity of European plant communities. I believe that showing only the positive sides of Robinia cultivation is very controversial and may arouse reluctance to this research among many readers. Robinia causes a strong unification of the phylogenetic structure of vegetation and floristic uniformity. Native species decline and are replaced with ruderal plants, see i.e.

 Piwczyński, M., Puchałka, R., & Ulrich, W. (2016). Influence of tree plan- tations on the phylogenetic structure of understory plant commu- nities. Forest Ecology and Management, 376, 231–237. https://doi. org/10.1016/j.foreco.2016.06.011  

Slabejová, D., Bacigál, T., Hegedüšová, K., Májeková, J., Medvecká, J., Mikulová, K., Šibíková, M., Škodová, I., Zaliberová, M., & Jarolímek, I. (2019). Comparison of the understory vegetation of native forests and adjacent Robinia pseudoacacia plantations in the Carpathian- Pannonian region. Forest Ecology and Management, 439, 28–40. https://doi.org/10.1016/j.foreco.2019.02.039

Vítková, M., Tonika, J., & Müllerová, J. (2015). Black locust – Successful invader of a wide range of soil conditions. Science of the Total Environment, 505, 315–328. https://doi.org/10.1016/j.scito tenv. 2014.09.104

Vítková, M., Sádlo, J., Roleček, J., Petřík, P., Sitzia, T., Müllerová, J., & Pyšek, P. (2020). Robinia pseudoacacia dominating vegetation types of Southern Europe: Species composition, history, distribution and management. Science of the Total Environment, 707, 134857. https:// doi.org/10.1016/j.scito tenv.2019.134857

The above studies show that sometimes more harm than good from the cultivation of black locust and its introduction should be limited only to justified cases, such as land undergoing remediation taking into account the problems related to the management of biological invasions, see i.e.

 Brundu, G., Pauchard, A., Pyšek, P., Pergl, J., Brunori, A., Canavan, S., Campagnaro, T., Celesti-Grapow, L., de Sá Dechoum, M., Dufour-Dror, J.-M., Essl, F., Flory, S. L., Genovesi, P., Guarino, F., Guangzhe, L., Hulme, P. E., Jäger, H., Kettle, C. J., Krumm, F., & Richardson, D. (2020). Global guidelines for the use of non-native trees: Awareness, sustainable use, invasion risk prevention and mitigation. NeoBiota, 61, 65–116. https://doi.org/10.3897/neobi ota.65.58380

Answer: As above-mentioned, both the Introduction and Discussion have been extensively revised to point out the disadvantages of black locust cultivation. Also, the part referring to the potential limitation of niches for black locust in southern Europe in a changing climate, have been included in the Introduction (see Ln 83-86). This aspect is no longer included in the Discussion.

L.302-345: Almost the entire Discussion page contains only three references to just two publications. It is simply an interpretation of the results with some conclusions, not a proper scientific discussion.

Answer: Pls see our comment above. The Discussion has been revised and enriched in references. Thank you for your enlightening recommendations.

The full scientific name of the species with authority Robinia pseudoacacia L. should only be given in the text when it is mentioned for the first time. The abbreviated version of R. pseudoacacia should be used each subsequent time.

Answer: The full scientific name with authority is given now only in the abstract (Ln 15) and when first mentioned in main text (Ln 72). Throughout the rest of the text, we use only the common name black locust.

Thank you very much for your valuable comments and suggestions that helped substantially in improving this manuscript

Round 2

Reviewer 2 Report

The authors responded to all my comments. The discussion was also significantly improved.